# The Growth Oscillator and Plant Stomata: An Open and Shut Case

**DOI:** 10.3390/plants12132531

**Published:** 2023-07-03

**Authors:** Derek T. A. Lamport

**Affiliations:** School of Life Sciences, University of Sussex, Brighton BN1 9QG, UK; derekt.t.a.lamport@googlemail.com

**Keywords:** growth oscillator, calcium signalling, arabinogalactan protein, pinball machine, stomata

## Abstract

Since Darwin’s “Power of Movement in Plants” the precise mechanism of oscillatory plant growth remains elusive. Hence the search continues for the hypothetical growth oscillator that regulates a huge range of growth phenomena ranging from circumnutation to pollen tube tip growth and stomatal movements. Oscillators are essentially simple devices with few components. A universal growth oscillator with only four major components became apparent recently with the discovery of a missing component, notably arabinogalactan glycoproteins (AGPs) that store dynamic Ca^2+^ at the cell surface. Demonstrably, auxin-activated proton pumps, AGPs, Ca^2+^ channels, and auxin efflux “PIN” proteins, embedded in the plasma membrane, combine to generate cytosolic Ca^2+^ oscillations that ultimately regulate oscillatory growth: Hechtian adhesion of the plasma membrane to the cell wall and auxin-activated proton pumps trigger the release of dynamic Ca^2+^ stored in periplasmic AGP monolayers. These four major components represent a molecular PINball machine a strong visual metaphor that also recognises auxin efflux “PIN” proteins as an essential component. Proton “pinballs” dissociate Ca^2+^ ions bound by paired glucuronic acid residues of AGP glycomodules, hence reassessing the role of proton pumps. It shifts the prevalent paradigm away from the recalcitrant “acid growth” theory that proposes direct action on cell wall properties, with an alternative explanation that connects proton pumps to Ca^2+^ signalling with dynamic Ca^2+^ storage by AGPs, auxin transport by auxin-efflux PIN proteins and Ca^2+^ channels. The extensive Ca^2+^ signalling literature of plants ignores arabinogalactan proteins (AGPs). Such scepticism leads us to reconsider the validity of the universal growth oscillator proposed here with some exceptions that involve marine plants and perhaps the most complex stress test, stomatal regulation.

## 1. Introduction

About four hundred million years ago, bryophytes evolved an upright sporophyte with a terminal sporogenous capsule that enhanced aerial spore dissemination but created the opposing demands of water retention while permitting carbon dioxide entry. The development of a waterproof cuticle saved water but restricted CO_2_ absorption; that was solved by the evolution of sporophyte stomata. Pairs of kidney-shaped guard cells form stomatal pores that respond to ambient conditions by opening and closing and thus allow a trade-off between water retention and CO_2_ entry. A critical balance depends on the osmotic generation of turgor pressure and the elasticity of guard cell walls with anisotropic reinforcement. Guard cell turgor depends on the regulated influx and efflux of specific osmolytes. Opening and closing are distinct processes; both depend on Ca^2+^ signalling. However, a simple model remains elusive. Even after 50 years, the extensive literature shows that major problems remain stubbornly unresolved. This implies that a major piece of the puzzle is missing. Certainly, the glaring omission of AGPs from the current discussion is a strong hint of their possible relevance, especially considering their location and Ca^2+^ binding properties. The following discussion outlines how AGPs may contribute to the Ca^2+^ homeostasis that underlies the mechanical dynamics of stomatal function. This may resolve the apparent paradox of Ca^2+^ required for both opening and closing stomata [1,2,3,4].

This brief personal historical perspective began in 1958 in Cambridge in the late fifties then the epicentre of biochemical energetics represented by David Keilin, Robin Hill, Peter Mitchell, A.F. Huxley, and Alan Hodgkin. Cambridge was also the epicentre of structural biochemistry spearheaded by Frances Crick’s elucidation of the DNA double helix, Fred Sanger’s breakthrough in sequencing protein (then DNA), and protein X-Ray diffraction by Perutz and Kendrew. Peter Mitchell’s discovery of chemiosmosis and the proton motif force generated by proton gradients across a membrane is one of the great Nobel Prize winners [5]. It paved the way to a profound understanding of biochemical energetics that powers all life on planet Earth. It solved the great biochemical mystery of ATP generation and beautifully integrates David Keilin’s electron transport chain. It was also apparent that while proton gradients in one direction generate ATP, in the reverse direction, ATP can pump protons out of the cell to establish a membrane electrostatic potential that enables cation import. Those were the two major roles of membrane proton pumps until the recent discovery of its third role, notably in Ca^2+^ homeostasis, which was quite unanticipated. The Plant Kingdom has recruited reverse proton gradients; a novel extrapolation of Mitchell’s proton pump integrates auxin-controlled growth with cell surface glycoproteins and Ca^2+^ signalling as a major regulator of plant growth (Figure 1). This contrasts with the Animal Kingdom where reverse proton gradients contribute mainly to indigestion and medication with proton pump inhibitors.

A Venn diagram: four components embedded in the plasma membrane generate cytosolic Ca^2+^. The Venn diagram emphasises common attributes shared by the four membrane components that are uniquely linked by their common function in Ca^2+^ homeostasis.
Auxin efflux PIN proteins transport auxin.Auxin activates proton pump to generate protons.Protons release Ca^2+^ from AGP-Ca^2+^.Open Ca^2+^ channels supply cytosolic Ca^2+^.

Cell surface arabinogalactan glycoproteins (AGPs) were for many years with only modest evidence [6] considered signalling proteins until comprehensive structural carbohydrate analyses of Li Tan and Marcia Kieliszewski at Ohio University elucidated a consensus structure [7] of the repetitive arabinogalactan glycomodule: A 3D-molecular modelling Eureka moment revealed AGP function in one fell swoop. Two glycomodule sidechains terminated by glucuronic acid residues were potential Ca^2+^-binding sites [8]. Further direct assay and molecular dynamics simulations confirmed that observation. Quite unlike Ca^2+^ stores of animal cells, plants locate their dynamic Ca^2+^ storage at the most strategic point of entry, the plasma membrane. The release of dynamic Ca^2+^ then identifies a novel function of proton pump activity.

Another centre of activity involved National Service at Number Two Radio School, RAF Yatesbury, where two years as an instructor in radio theory, with a focus on oscillators, provided the essential backdrop to biological oscillators. This knowledge was later critical in identifying the AGP-Ca^2+^-dependent Ca^2+^ oscillator [8]. Over 25 years ago, oscillatory growth was correlated with oscillations in the pollen tube tip focussed Ca^2+^ gradient; this suggested Ca^2+^ influx could originate from external stores in the cell wall [9]. However, AGPs at the pollen tube tip [10] became significant when the recent discovery of AGP-Ca^2+^ and the cell surface Ca^2+^ oscillator explained its crucial role in pollen tube growth. The wide ramifications include not only tip growth of pollen tubes but numerous other plant processes such as phyllotaxis and even stomatal movements [11].

To some, these ramifications may appear contentious. They merit further consideration. How does this simple pinball hypothesis involve such profound interactions? Mitchell’s proton pump and ATP generation drive virtually all biochemical processes. In both plants and animals, a dual role of proton pumps generates ATP and also the negative potential that drives cation influx across the plasma membrane. However, a third major role unique to plants recruits proton pumps that generate reverse proton gradients to enable Ca^2+^ homeostasis that regulates plant growth and differentiation thus exemplifying the parsimony of nature (Figure 1).

These broad generalisations might appear as extravagant claims. The following sections outline the historical background and discuss the pro and cons of the pinball hypothesis particularly in the light of recent papers. It concludes with stomatal regulation as a stress test of stomatal dynamics as the most complex example:

### 1.1. Historical Background

Sydney Ringer’s serendipitous discovery of Ca^2+^-dependent muscle contraction [12] was followed somewhat later by the identification of Ca^2+^ as essential for pollen germination [13]. These discoveries stimulated huge interest in both animals and plants and generated prolific literature over many years. Animal cells have two sources of dynamic Ca^2+^, a plentiful external supply and a more limited internal store within the endoplasmic reticulum (ER) released to the cytosol by inositol triphosphate IP3 of the Ca^2+^-signalling phosphoinositide (PI) pathway [14]. Many consider that this pathway also operates in plants [15] reminiscent of T.H. Huxley’s famous but misleading aphorism that “A plant cell is an animal in a wooden box”. However, profound differences between animal and plant cells include AGPs unique to plants and the missing link in Ca^2+^ homeostasis: Indeed, it is now clear that plants store dynamic Ca^2+^ externally as periplasmic AGP-Ca^2+^ rather than in the ER. One hardly expects immediate acceptance of this model. Time sifts. Paradigms shift. Indeed our inference extrapolates Peter Mitchell’s chemiosmotic model that couples proton gradients across mitochondrial membranes with ATP generation [5] (Nobel Prize 1978). Across plasma membranes, such gradients also involve proton secretion. Plants have uniquely recruited that property to regulate Ca^2+^ homeostasis and growth. Somehow, plants have bypassed the enormous complexities of hormonal regulation in animals with an elegant strategy that regulates plant growth by unifying an auxin-activated proton pump with AGPs and Ca^2+^ homeostasis. 

### 1.2. Simplicity

A friendly critic ignoring Occam’s razor considers the four-component oscillator “too simple”. However, each component has many forms, notably 11 proton pumps in Arabidopsis AHA1 to AHA11, and 9 PIN auxin efflux proteins that solve the long-standing problem of polar auxin transport. Together with multiple Ca^2+^ channels [16] and AGPs the oscillator is superbly adaptable.

### 1.3. The Mathematical Basis

Previous attempts to impose mathematical rules on plant growth [17] lack the essential biochemical details described here. Although one might expect some mathematical insight, the detailed structure of the crucial AGP glycomodule with paired glucuronic acid residues based on impeccable NMR analyses [18] and molecular simulations of Ca^2+^-binding [8] all depends on sophisticated mathematical techniques. 

### 1.4. Cytosolic Ca^2+^: Is AGP Glucuronic Acid the Source?

It is generally assumed that Ca^2+^ freely available in the plant apoplast is the immediate source of dynamic cytosolic Ca^2+^ as well as internal stores [15] thus similar to the classical animal model [14] that involves storage in the ER and vacuole with release mediated by inositol triphosphate (IP3), but fundamentally quite different from the model described here which is specific to plants. 

The crucial role of the proton pump in generating dynamic cytosolic Ca^2+^ in plant cells is clearly evidenced by pollen tube tip-focussed Ca^2+^ influx, summarised by >30 years work of [19]. However, despite the evidence of AGP periplasmic location [20] and detailed structure of AGP glycomodules, many papers ignore the proposed role of AGPs as a precursor of cytosolic Ca^2+^ facilitated by paired sidechains with terminal glucuronic acid residues that enable stoichiometric Ca^2+^-binding essential to the Ca^2+^ storage function of AGPs.

Every model has a weak link: Are glucuronic acid residues essential? Theory predicts but experiment decides: The 2:1 glucuronic: Ca^2+^ stoichiometry [8] suggests a critical test by generating classical AGPs that lack glucuronic acid. Although only a “minor” component, glucuronic acid is invariably present in classical AGPs, but more abundant in AGPs of plants such as Eelgrass and monocot *Zostera marina* that have returned to the high salt levels of a marine environment [21]. Presumably, the increased glucuronic acid content is an evolutionary adaptation that enhances Ca ^2+^ binding, signifying a pivotal ecological role for AGPs [22].

However, multiple glucuronosyl transferases (GlcATs) of the Arabidopsis genome create significant genetic redundancy and therefore a challenge to functional analysis.

### 1.5. GlcAT Knockouts Affect Ca^2+^ Homeostasis

A critical test of the AGP-Ca^2+^ hypothesis by two groups with substantial contributions to AGP structural chemistry generated multiple mutants that lack AGP glucuronic acid, the Dupree group at Cambridge UK, and the Showalter group at Athens Ohio. The genetic redundancy of glucuronosyl transferases that add GLcA to AGP polysaccharides involved complementary approaches to the generation of multiple GlcAT gene knockouts: Firstly, T-DNA insertion lines involving Ti plasmid transformation mediated by *Agrobacterium tumefaciens* generated Arabidopsis mutants in four glucuronosyl transferases (GlcAT14A, -B, -D, and -E) [23]. 

Secondly, similar Arabidopsis mutants were generated by the CRISPRCas9 multiplexing approach for GlcAT14A, GlcAT14B, and GlcAT14C [24]. 

A comprehensive and detailed study of plant growth by the Dupree group showed that multiple glucuronosyl knockouts severely impaired Ca^2+^ homeostasis. This was consistent with the greatly decreased glucuronidation and Ca^2+^-binding capacity of AGPs isolated from leaves and roots of glcat14a/b/d mutants [23], while the growth of the glcat14a/b/e triple mutant was poor. Other mutants showed numerous pleiotropic growth effects: shorter plants, decreased leaf expansion (including cell shape formation and expansion), shorter hypocotyls lacking an apical hook, trichomes with decreased branching, and abnormal Ca^2+^ transients in roots with attenuated Ca^2+^ wave propagation. Significantly, exogenous Ca^2+^ suppressed/rescued these AGP glucuronidation mutant phenotypes.

A parallel study by the Showalter group focussed on sexual reproduction and the role of multiple glucuronosyl transferase (glcat) mutants in Arabidopsis. Over many years significant contributions to AGP biology by the Showalter group have demonstrated the widespread involvement of AGPs encoded by 85 genes and 11 GLCAT genes. Their current work demonstrates that multiple glcat mutants significantly impair sexual reproduction, particularly pollen development, polytube block, and successful fertilisation that requires cytoplasmic calcium oscillations in synergid cells after physical contact with the pollen tube tip followed by normal embryo development in Arabidopsis.

Taken together, the Dupree and Showalter groups corroborate the pinball model and support a unified theory of plant growth based on the properties of membrane glycoproteins.

Multiple glucuronosyl transferase knockouts severely impair Ca^2+^ homeostasis, therefore Glucuronic-bound Ca^2+^ is the essential source of cytosolic Ca^2+^. The novel idea of a Ca^2+^ store strategically located at the surface of the plant plasma membrane was previously overlooked because Mother Nature keeps some of her best secrets well hidden.

### 1.6. Is the Pinball Machine Random?

In a conventional pinball machine, pinball trajectories are largely random. Thus, although a strong visual metaphor a molecular pinball machine (Figure 1) has limitations but clear advantages: All four are components of the plasma membrane that emphasise their close physical proximity, hence a single entity integral to their overall function. Indeed, while the trajectory of physical pinballs may appear random, the plasma membrane is far from random consisting of a highly organised hydrophobic lipid surface covered by a hydrophilic glycoprotein layer of AGPs [20]. At such a hydrophobic–hydrophilic interface, nanoconfined water may behave as an interfacial layer where protons diffuse within a matrix with high ionic conductivity that occurs through surface hopping of protons [25]. Thus proton conduction connects the proton pump with AGP glycomodules and may explain very rapid responses. The glycomodule structure itself may facilitate proton conduction to the Ca^2+^-binding centre.

### 1.7. What Connects Auxin with Ca^2+^ Signalling?

Although a connection between auxin and Ca^2+^ has been apparent for many years [26], both auxin and fusicoccin activate the proton pump [27] with a concomitant rapid elevation of cytosolic Ca^2+^. While Vanneste and Friml concluded [28] that “the role of auxin-induced Ca^2+^ signalling is poorly understood”, Dindas and colleagues confirmed [29] a close connection, now described by the pinball model, where periplasmic AGPs are a sine qua non of Ca^2+^-mediated metabolic regulation. An alternative explanation postulates complex auxin-activated Ca^2+^ channels [30]: Auxin (IAA) enters cells through the protein AUX1 and binds to a protein of the TIR1/AFB family. This leads to rapid ion flow (within seconds) through the protein AHA and a Ca^2+^ channel. While they demonstrate Ca^2+^ influx within seconds of IAA application, it completely ignores the source of dynamic Ca^2+^ bound by cell surface AGP-Ca^2+^ stores, hence overlooking the direct role of the proton pump. 

### 1.8. Origin of Growth Oscillations and Ion Fluxes

Oscillations in growth and cytosolic Ca^2+^ fluxes of pollen tube tips [19,31] and root hairs [32] suggest a fundamental growth oscillator; but what activates the oscillator to initiate these oscillations? The pinball model identifies the proton pump as a prime candidate triggered by auxin and mechanostress. While Ca^2+^ oscillations have been described as inherent in root hairs and pollen tubes [31] ubiquitous auxin efflux “PIN” proteins transport auxin throughout the plant [33,34].

### 1.9. Stomata Test the Pinball Hypothesis

Proton pumps drive a pinball machine that generates cytosolic Ca^2+^.

Auxin is the accelerator and abscisic acid is the brake:

In hydrated cells, an auxin-activated pump dissociates AGP-Ca^2+^ that triggers rapid Ca^2+^ oscillations thus increasing osmolytes K^+^, malate, and turgor.

In stressed cells, abscisic acid inactivates the proton pump thus decreasing Ca^2+^ oscillations, osmolytes, and turgor. 

A contribution to current descriptions of stomatal regulation runs the risk of oversimplification (Figure 2). Balancing opposing demands of water conservation and CO_2_ uptake involves a complex guard cell wall structure and numerous signals mediating subtle control of osmolytes and turgor pressure. They include the Ca^2+^ signal and its oscillations discussed in a recent comprehensive review [35] that assumes the source of cytosolic Ca^2+^ is largely from endomembrane stores, now almost an article of faith, although at the growing tip of the pollen tube, the cytosolic Ca^2+^ source is quite clearly external. Nevertheless, “Best estimates indicate that endomembrane release accounts for more than 95% of the Ca^2+^ entering the cytosol” [4]. That seems inaccurate as it ignores AGPs the missing link connecting AGP-Ca^2+^ with the proton pump whose primary role maintains a large negative membrane potential essential for stomatal opening via influx of K^+^ osmolyte and concomitant malate synthesis that increase turgor pressure. Significantly, treatment with auxin and fusicoccin also opens stomata similar to the constitutively fully active proton pump mutant AHA1 [3].

Arguably, the proton pump regulates Ca^2+^ homeostasis. This is a more subtle role than maintaining membrane potential but of hitherto unappreciated significance. Thus, immunolocalisation of AGPs and detection of their abundance in guard cells by the Yariv reagent [36] confirm Ca^2+^ involvement in stomatal regulation. However, precisely how plants achieve specificity in intracellular calcium signalling is largely speculative [4]. Some suggest [3] that “Oscillations in cytosolic Ca^2+^ allow for information to be encoded in both the amplitude and frequency”. This “calcium code” has spawned a voluminous literature dealing with stomatal closure and associated Ca^2+^ oscillations; their frequency and amplitude trigger opening and closure by cation osmolyte influx and efflux, respectively. Not surprisingly, metabolic activity, highest in turgid cells, triggers the highest cytosolic Ca^2+^ oscillations [2]. In open stomata, the high malate concentration (25 µM cytosol; 464 µM vacuole [1]) is of particular significance as malate chelates excess Ca^2+^ that might otherwise overwhelm the cytosol. 

The proton pump acts as a central hub regulated by virtually all factors known to regulate plant growth, hormones, and classical plant hormones [37] generating the complexity summarised by [38].

Figure 2 compares turgid and flaccid stomatal guard cells:

In turgid guard cells, the proton pump activated by auxin releases Ca^2+^ from AGP-Ca^2+^ and also generates the membrane potential. Thus AGP-Ca^2+^ dissociation initiates cytosolic Ca^2+^ oscillations while a large membrane potential enables K^+^ influx increasing cytosolic osmolyte levels including malate derived from starch. These Ca^2+^ oscillations provide the cytosolic Ca^2+^ that triggers vesicle exocytosis [39,40]. Indeed, the pivotal metabolic role of Ca^2+^ is evidenced by the over 200 EF-hand Ca^2+^-binding proteins as key transducers mediating Ca^2+^ action encoded in the *Arabidopsis thaliana* genome [41]. 

In flaccid cells, osmotic stress and dehydration generate abscisic acid, which triggers stomatal closure [42] by inactivating proton pump activity [43]. This results in membrane depolarisation that enables guard cells to jettison K^+^ osmolyte via the specific K^+^ efflux channel GORK [44] and recycle malate as starch. Thus, the transition between the turgid and flaccid cell states depends on Ca^2+^ oscillations that regulate osmolyte influx and efflux respectively; increased cytosolic Ca^2+^ opens stomata [2,26,45]. On the other hand, the flaccid guard cells of closed stomata are virtually quiescent with low metabolic demands just sufficient to open K^+^ efflux channels and convert malate to starch. Such low demands presumably reflect the much weaker Ca^2+^ oscillations hence a low level of metabolic activity. This resolves the paradox, depicted in Figure 2, which represents a simplified scenario. 

## 2. Ecological Significance of AGPs

The coastal town of Brighton afforded access to a wide range of ecological habitats including chalk downland, freshwater, estuarine, marine, and saltmarsh environments. These included the Lewes Brooks where salt ingress from the adjacent tidal river Ouse formed a clear ecological gradient. Figure 3 shows two schoolboys with their biology teacher circa 1950, titrating the salt levels of a brook where salinity decreased with increasing distance from the river.

These early influences, which included summers living with a farm family and working on a mixed dairy/arable farm, led to a curiosity about natural processes and a career in biochemistry. On retirement joining the faculty at Sussex University as a visiting professor, Dean Timothy Flower’s interest in saltmarsh ecology stimulated my approach to salt stress and led to the discovery of upregulated AGP secretion in cultured cells [20]. Elucidation of a consensus structure for the complex AGP polysaccharide by Li Tan and Marcia Kieliszewski at Ohio University [7], enabled molecular modelling that identified paired glucuronic acid residues hence the significant Ca^2+^ binding role of classical AGPs [8] that led to the novel concept Ca^2+^ homeostasis regulated by a molecular pinball machine in the plasma membrane [46,47,48,49]. Possible exceptions to AGP-Ca^2+^ binding were of great interest, particularly in seawater species of flowering plants where high Na^+^ could disrupt Ca^2+^ binding. This was an evolutionary conundrum faced by flowering plants seeking to escape drought through the return to an oceanic abundance. Indeed, evolutionary adaptation to high salt has occurred in a family of seagrasses represented by *Zostera marina* [21,22]. It creates sea meadows that form large well-defined habitats supporting a diverse community of vertebrates including sea horses and invertebrate microscopic crustacean Copepods that play an essential role in Zostera reproductive biology. With the complete absence of terrestrial insect pollinators, Copepods have stepped into the breach by feeding on Zostera pollen ensuring cross-pollination. Thus, microscopic Copepods enable an ecosystem that acts as a massive contributor to carbon capture and photosynthetic oxygen evolution worldwide. While Zostera’s adaptation evolved relatively recently (<20 MYA), the more ancient chlorophycean alga *Ulva lactuca* is also adapted to a marine environment but 90% of Chlorophyta are freshwater species; thus Ulva probably migrated from fresh to seawater very much earlier than Zostera [21], yet quite remarkably shows similarly adapted AGPs with increased glucuronic acid content and Ca^2+^-binding [50]. Thus it is a fine example of convergent molecular evolution even though separated by many millions of years.

While highly glycosylated classical AGPs are the most abundant in cell extracts, a small minority of AGPs with just a few arabinogalactan glycomodules are designated as non-classical; such minimal glycosylation directs them into the Golgi secretion pathway targeted to the plasma membrane [51]. Their functional elucidation presents a challenge for present and future plant scientists.

## 3. Conclusions

This review recapitulates recent significant progress in AGP biology represented by the molecular pinball machine of the plasma membrane that unifies proton pumps, AGPs, and Ca^2+^ signalling [46].

Here, we presented a critical test of the pinball hypothesis and suggested how it resolves the paradox that Ca^2+^ can both open and close stomata, with AGPs as the missing link and source of cytosolic Ca^2+^. Arguably, the proton motive force controls both the opening and closing of stomata. Thus positive and negative regulation of the proton pump involves an auxin accelerator and an abscisic acid brake. That simple analogy explains how low Ca^2+^ levels close stomata, emphasising elegant control of both water loss and carbon dioxide gain.

Recent work validates the pinball hypothesis and is a suitable closure to a research career extending over 60 years initiated by the discovery of hydroxyproline-rich cell wall proteins [47] in 1960. Plants have uniquely recruited hydroxyproline-rich proteins, extensins, AGPS, and their allies, to serve dynamic functions that range from direct involvement in cell division [48] and gravitropism [49] to Ca^2+^ homeostasis and its related tropisms. It has taken many years to appreciate the versatility of these cell surface glycoproteins.

## Figures and Tables

**Figure 1 plants-12-02531-f001:**
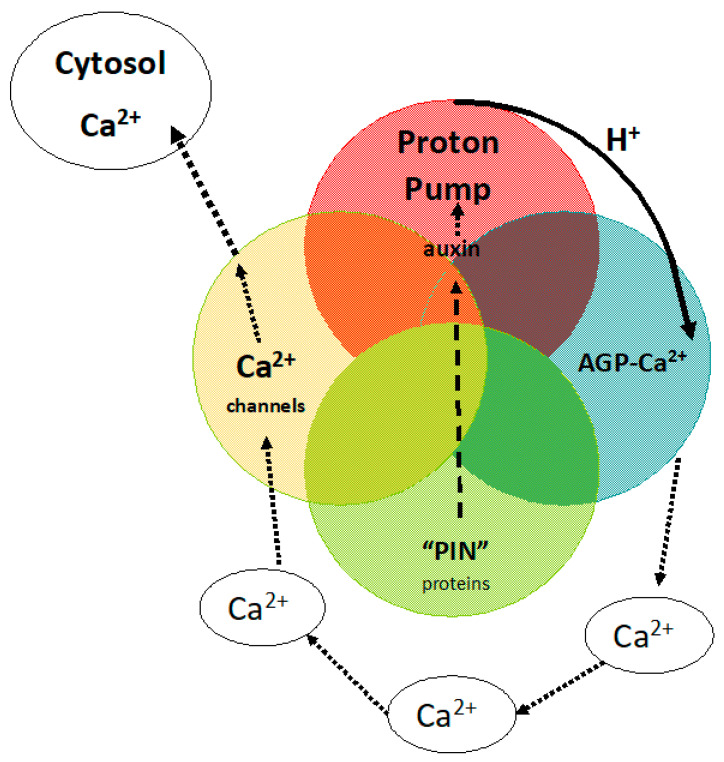
The pinball machine encapsulates the role of the proton pump in Ca^2+^ homeostasis.

**Figure 2 plants-12-02531-f002:**
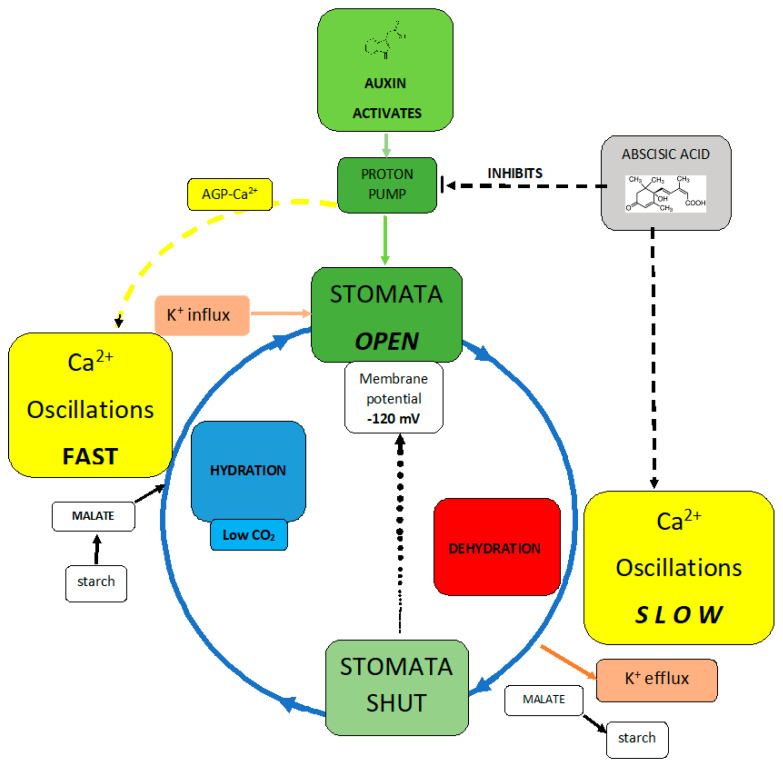
Stomatal dynamics summarised.

**Figure 3 plants-12-02531-f003:**
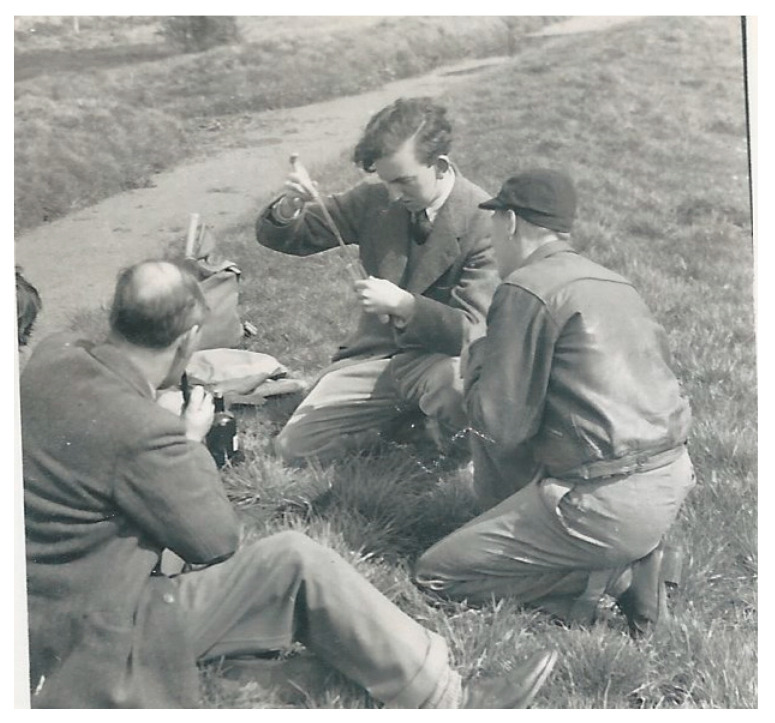
Titrating salt levels in the Lewes Brooks next to the River Ouse, near Lewes, East Sussex, UK. From left to right, Douglas Pratt, Biology teacher, with schoolboys A.V. Grimstone and Derek T.A. Lamport from Varndean School for boys sixth form, Brighton [Photo dtal].

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
