# Peer review of "The Growth Oscillator and Plant Stomata: An Open and Shut Case"

_plants, 2023, doi:10.3390/plants12132531_

Round 1
Reviewer 1 Report
In this manuscript, D. Lamport presents a comprehensive review of the literature concerning the role of a fascinating system comprising plasma membrane-localized arabinogalactan glycoproteins, Ca2+, proton pumps, and auxin in regulating the movements of very important specialized cells, stomata guard cells.
The text is well-written and clear. The two figures are well presented.
I have only very minor points to raise, as there was no line number indicated in the submitted manuscript, I will provide the page number and the sentences to be corrected.
Page 2 “figure 1. ” the figures are referred to as “Figure 1.” Elsewhere in the text.
Page 4 bottom, “AGPS” should be “AGPs”
Page 5 part 4 “It is generally assumed that Ca2+ freely available in the in the plant” delete one “in the”
Page 6 “Agrobacterium tumefaciens” should be in italics
Reference list: several points to modify
Ref 3 format
Ref 12 delete Ref ID: 13033
Ref 38 format
Ref 48 format
Ref 49 delete Ref ID
The review addresses the question of the regulation of stomata functioning in plants, and in particular the gap and paradox that Ca2+ is involved in both opening and closure of stomata. The topic is original and important. Plant stomata are crucial cellular structures that regulate gas and water exchange with the environment. Not adapted, the manuscript is a review not an article with experimental work. They cover the field and include initial references and very recent articles. However, as indicated in my initial report, the format of some references in the list needs to be corrected. The two figures are well presented and illustrate the models explaining in particular how auxin, proton pumps, AGP and Ca2+ are involved in the stomata opening and closure.
Author Response
Many thanks to this reviewer for attention to the fine details and kind words . I have made the corrections suggested.
Reviewer 2 Report
The paper summarizes the relevant research to date on Ca2+ oscillator, a molecular PINball machine, which functions through the combination of auxin-activated proton pumps, AGPs, Ca2+ channels, and auxin efflux “PIN” proteins embedded in the plasma membrane, with historical background. It also discusses the pro and cons of the pinball hypothesis (AGP as a dynamic Ca2+ storage) consisting of those four components, and finally introduces the role of Ca2+ oscillator in an open and shut stomata when AGP is the source of cytosolic Ca2+.
There are two main themes of this paper: (1) The history of research related to the pinball hypothesis and its pro and cons: (2) Adaptation of the pinball hypothesis in Stomata. The author has written several review articles in recent years including theme (1), but the present article differs from them in that it summarizes a series of studies without omission, from the discovery of the proton motif force to the discovery of AGPs as sources of a dynamic Ca2+ and the latest experiments to test the pinball hypothesis. What makes this paper particularly unique is the theme (2). The structure of the paper is such that the sections related to (1) and the last section on (2) are described in parallel, but it might be better to separate (1) and (2) in a larger section.
The following minor modifications could be made.
In the section on the “simplicity” of the pinball hypothesis, the critical opinions of critics are humorously described using Occam's razor, if references to those critical opinions exist, they should be cited.
Figure 1 depicts a Venn diagram that briefly illustrates the pinball hypothesis, which consists of four elements on the membrane. The pathways indicated by the arrows are clear, but does the overlap in the Venn diagram mean anything? If there is an intention, it should be described in the legend.
The pathways indicated by the arrows are clear in both Figs. 1 and 2, but the distinction between inside and outside the cell is not illustrated and difficult to understand. It might be possible to draw a diagram showing the plasma membrane, as in Fig. 1, Derek T. A. Lamport et al., 2020 “Phyllotaxis Turns Over a New Leaf—A New Hypothesis”, in both open and shut stomata respectively.
Author Response
Many thanks to this reviewer for the stimulating questions and suggestions.
Providing larger sections for (1) or (2) would be rather too difficult at this stage and would require a substantial effort. Section (2) is shorter for several reasons. Stomatal Ca2+ was the initial stimulus because it seemed like a good test of the pinball hypothesis and helped to focus the discussion.
The pinball hypothesis was described by my colleague Keith Roberts as “too simple” in response to my earlier email. I have simply turned that criticism around by citing Occam’s Razor so readers can make up their own mind! Admittedly it is a very simple depiction that summarises exceedingly complex molecular phenomena.
The Venn diagram emphasises the common attributes shared by four membrane components that are uniquely linked by their common function in Ca2+ homeostasis, so I have added this brief explanation to the legend.
Reviewer 3 Report
This manuscript was fun to read, although it is somewhat out of my field of expertise, so I had to read several of the references to orient myself.
My only concern is that an oscillator causes a rise and fall in the signal. The author only describes the rise in cytoplasmic Ca++, but not why it would fall.
I also think it would be good to give an idea of the actual time frames of the oscillations. Some seem to be diurnal and others a matter of minutes.
What proportion of all AGPs have the paired glucuronic acids? How thoroughly has this been investigated?
Author Response
Thanks to this reviewer for helpful comments.
Cytosolic Ca2+ levels fall for several reasons; these include rapid sequestration by vacuole and ER but also recycling via membrane Ca2+ ATPase and AGP export.
Oscillation time frames are mentioned in many papers, such as re 2 and ref 3, and exemplified by abscisic acid that “dampens” these oscillations. The pollen tube is the best example of rapid oscillations with a period of about 30 seconds.
The proportion of AGPs with paired glucuronic acids judged from the GlcA:Ca2+ stoichiometry of 2:1 is large judging from my own assays of Ca2+ binding by AGPs, and corroborated by the excellent work with GlcA transferase knockouts from the Dupree and Showalter labs.
I noted an additional point of great current interest, notably marine grasses like Zostera which have enhanced AGP glucuronic acid content as an adaptation to high salt levels. These monocots are great carbon sequesters and form ecologically important sea meadows. They piqued my interest because they flower just like terrestrial angiosperms with the exception that insect pollinators are replaced by Copepods, microscopic crustaceans that feed on the pollen and thus enable cross-fertilisation!